# Three-Dimensional Bio-Printed Cardiac Patch for Sustained Delivery of Extracellular Vesicles from the Interface

**DOI:** 10.3390/gels8120769

**Published:** 2022-11-25

**Authors:** Assaf Bar, Olga Kryukov, Smadar Cohen

**Affiliations:** 1The Avram and Stella Goldstein-Goren Department of Biotechnology Engineering, Ben-Gurion University of the Negev, Beer-Sheva 84105, Israel; 2Regenerative Medicine and Stem Cell (RMSC) Research Center, Ben-Gurion University of the Negev, Beer-Sheva 84105, Israel; 3The Ilse Katz Institute for Nanoscale Science and Technology, Ben-Gurion University of the Negev, Beer-Sheva 84105, Israel

**Keywords:** cardiac patch, extracellular vesicles, 3D bioprinting, alginate sulfate

## Abstract

Cardiac tissue engineering has emerged as a promising strategy to treat infarcted cardiac tissues by replacing the injured region with an ex vivo fabricated functional cardiac patch. Nevertheless, integration of the transplanted patch with the host tissue is still a burden, limiting its clinical application. Here, a bi-functional, 3D bio-printed cardiac patch (CP) design is proposed, composed of a cell-laden compartment at its core and an extracellular vesicle (EV)-laden compartment at its shell for better integration of the CP with the host tissue. Alginate-based bioink solutions were developed for each compartment and characterized rheologically, examined for printability and their effect on residing cells or EVs. The resulting 3D bio-printed CP was examined for its mechanical stiffness, showing an elastic modulus between 4–5 kPa at day 1 post-printing, suitable for transplantation. Affinity binding of EVs to alginate sulfate (AlgS) was validated, exhibiting dissociation constant values similar to those of EVs with heparin. The incorporation of AlgS-EVs complexes within the shell bioink sustained EV release from the CP, with 88% cumulative release compared with 92% without AlgS by day 4. AlgS also prolonged the release profile by an additional 2 days, lasting 11 days overall. This CP design comprises great potential at promoting more efficient patch assimilation with the host.

## 1. Introduction

Heart failure (HF) is considered a major public health concern, affecting millions of people worldwide [1]. HF is frequently a complication of a myocardial infarction (MI), outlined by massive cardiac muscle death due to ischemia [2]. The poor intrinsic regenerative capacity of the adult mammalian heart comprises the main burden limiting recovery from myocardial injuries. In the last three decades, cardiac tissue engineering (TE) has emerged as a therapeutic approach, suggesting the use of biomaterials, mostly in combination with bioactive molecules and/or a cell source for regeneration [3].

In recent years, it was suggested that cardiac regeneration is mainly propelled by paracrine signaling, giving rise to an opposing strategy, based mostly on the administration of cell-secreted, bioactive signaling molecules (i.e., growth factors and miRNA) [4,5]. Representatives of these types of inducers are extracellular vesicles (EVs), which are small, natural membrane-enclosed carriers, generated and secreted by living cells [6]. It has been established that EVs are used to mediate cell-to-cell communication [7] and their content usually includes proteins, mRNA and miRNA. Prominent examples can be found in cancer cells [8,9] and in the cardiovascular system [10]. EVs from various cell types were shown to have potential to improve cardiac repair by promoting angiogenesis [11], inflammation and oxidative stress reduction [12,13] and stimulation of cardioprotective-associated cell processes [14], making them potential vehicles for drug delivery. Even though an EV-based therapeutics presents major advantages, including specific cellular uptake and it is considered to be non-toxic, its short-term effect is still a limitation. Similar to cytokines, circulating EVs have a short half-life and are internalized fast by their recipient cells, therefore restricting their impact to a narrow time window [15].

Drug delivery platforms, allowing controlled and sustained release of bioactive molecules, can be fabricated from biomaterials. Their mechanical properties can be customized by applying different fabrication methods and by chemical formulation and modifications [16,17]. Therefore, such matrices can present and deliver drugs through various mechanisms, depending on their degradation/erosion rate, exogenous triggers or their surrounding conditions [18]. For instance, Liu et al. introduced hydrogel microcapsules, fabricated from natural polymer composites using microfluidics to produce nanomedicine. The microcapsules were capable of sustained release dependent on surrounding pH and temperature, suitable for oral administration [19]. One group of biomaterials widely investigated for drug delivery applications are natural polymers, including chitosan [20], hyaluronic acid [21], collagen [22] and alginate [23]. The latter was vastly investigated for TE applications as injectable hydrogel [24,25,26] or as a scaffold [27,28,29]. In itself, alginate is not known to be biologically active, as it lacks molecular recognition [30]. Our group developed an alginate-based delivery system, enabling the presentation and controlled release of multiple growth factors via affinity binding interactions to alginate sulfate (AlgS) [31]. The latter mimics in large the interactions of growth factors, chemokines, and cell adhesion molecules, collectively known as heparin-binding proteins with heparin and heparan sulfate. The alginate–AlgS system was demonstrated in chondrogenic, cardiac and ovarian models, allowing sustained release and prolonging the beneficial effect of heparin-binding growth factors in vitro and in vivo [32,33,34]. 

The ultimate destination of Cardiac TE is the assembly of a fully developed, three-dimensional (3D) functioning heart tissue, cultivated ex vivo, available for transplant. In cases of cardiac malfunction, such patches can be used as substitutes replacing large fractions of the injured heart [35,36,37]. Many cardiac patches (CPs) designs are focused on providing engrafted cells with a suitable microenvironment for proper development and maturation of the cardiac tissue. Such designs have demonstrated improved cell retention and feasibility to achieve a clinically relevant cardiac graft in means of size and function [38,39]. More recently, 3D bio-printing has materialized for fabrication of CPs for multiple purposes, from improving cell delivery and retention [40], through spatial distribution of endothelial cells for improved vascularization [41], to drug delivery applications [42]. 

However, to successfully employ CPs as a therapeutic strategy, some restrains still need to be overcome. To truly restore cardiac function, the CPs must integrate properly with the surrounding myocardium at three levels: physical and biochemical continuum, electrophysiological coupling, and nutrient supply. For instance, the designed patches must be accessible for cells to infiltrate and migrate from regions close to the infarcted zones and enable blood vessels and nerve formation [43,44]. In addition, it is mandatory that these networks will integrate accordingly with those of the host. Furthermore, a graft lacking both mechanical and electrical coupling with the host, could result in arrhythmias caused by unsynchronized electrophysiological signal transmission between the graft, the host myocardium and the fibrotic interface in between [45]. Chung et al. addressed this problem in a study performed on primate hearts, using hESC-CM transplantation, as ventricular arrhythmias were detected consistently, differently from previous observations in smaller animal models [46]. These events were attributed to the large distances the electrical and mechanical signals must travel through the tissue, thus decreasing conductivity. In small animal models, the presence of a fibrotic scar, acting as a barrier for electrical signal transduction led to a failing electrical integration of the CP with the host [47]. Therefore, induction of regenerative processes specifically at the interface between the patch and the host is of great significance. 

To address these issues, we suggest herein a novel, 3D bio-printed cardiac patch design, composed of two compartments; an inner cell-laden compartment, enveloped by an outer compartment containing EVs. It is postulated that controlled, sustained delivery of EVs through release from alginate–AlgS matrix, would promote cardiac regeneration and improve integration if applied locally at the interface between the patch and surrounding tissue.

## 2. Results

### 2.1. Cardiac Patch Design

The proposed design for the CP includes two main compartments (Figure 1A). The inner core is composed of a RGD-modified alginate (Alg-RGD) composite matrix and cells. The outer shell (400 µm thick, symmetrical from all sides) serves as a delivery system, based on alginate hydrogel impregnated with EVs, affinity-bound to alginate sulfate (AlgS), enabling the controlled release of EVs from the matrix by two mechanisms of action: (a) erosion of the alginate matrix and (b) affinity binding of the EVs to alginate sulfate. The delivery compartment is designed to act mainly at the interface of the CP with host myocardium after transplantation.

For CP fabrication, we approached 3D bio-printing of hydrogels for two main reasons: (a) specific localization of EVs within the shell of alginate construct, acting as a drug delivery system combined with tissue replacement and (b) Controlling scaffold architecture on both macro and micro scale. 

### 2.2. Rheological Properties and 3D Bioprinting of Alginate-Based Solutions

For successful 3D bio-printing of hydrogels, a key requirement is the bioink’s capability to undergo liquid-to-gel phase transition. First, the bioink should be injectable to be applied to the surface. On the other hand, it must form a gel rapidly enough as it meets the printing surface. Since the mechanism of alginate hydrogels’ crosslinking is based on electrostatic interactions mediated by calcium ions, the latter indicated under which conditions phase-transition takes place. Furthermore, adding components which disturb the degree of crosslinking (i.e., AlgS, EVs and\or cells) also affects the hydrogel’s mechanical properties. In the proposed system, two types of hydrogel formulation are required—one for the inner core and one for EV-containing shell. Therefore, the effect of inserted components on each of the bioink solutions mechanical properties was evaluated.

#### 2.2.1. Cell-Laden Core Bioink Solutions

For fabrication of the inner core, three formulations of bioink solutions were examined. All tested bioink solutions were composed of calcium crosslinked Alg-RGD, to provide the cells inside the matrix with the biochemical cues required for cell–matrix interactions. However, the bioink solutions differ from one another by their polymer composition and ratios, specified in Table 1. Generally, since 3D bio-printing was performed using the FRESH approach using gelatin beads washed with 0.1 M CaCl_2_, the printing temperature had to be set between 25–30 °C. Otherwise, the solution would either block the printing nozzle (below 25 °C) or solubilize the gelatin beads (30 °C or above), which both lead to poor 3D printing outcome. 

The first bioink solution tested contained only crosslinked Alg-RGD (1.7% Alg-RGD, 0.36% Ca^2+^, Figure 2, left) and the first parameter examined was whether the inclusion of cells in the bioink affects the viscoelastic behavior of the bioink solution. Cell addition to the bioink slightly shifted the location of the cross point between storage modulus G′ (elastic response) and the loss modulus G′′ (viscous response), however did not alter its viscosity significantly. In general, both the cell-laden bioink and bioink without cells presented liquid-like mechanical properties at low angular frequency (Figure 2A, left), making it applicable for 3D bio-printing. Nevertheless, even though the FRESH gelatin bead contains 0.1 M CaCl_2_ to further crosslink the 3D bio-printed constructs during fabrication, liquid-like biomaterials are more susceptible to gradual dissociation over time, resulting in less mechanically stable constructs.

Since 3D construct stability is important in the fabrication of a CP (i.e., for patch retention over a beating heart), we speculated that increasing of the bioink solution’s viscosity, presenting more gel-like behavior, would produce better results in terms of mechanical stability post-printing. However, even a slight increase in calcium crosslinking degree (over 0.36%, *w*/*v*) might result in a non-printable hydrogel. Moreover, a higher degree of crosslinking could be harmful for encapsulated cells, resulting in nanoscale pore size of the resulting hydrogel, affecting both mass transfer within the printed strands and obtained 3D cell morphology. Therefore, it was decided to add gelatin to the bioink solution to obtain increased viscosity during 3D bio-printing (Table 1, bioink 2). Another incentive for gelatin addition was its solubility at 37 °C, similar to the process occurring in FRESH solubilization post-printing. Hence, in the case of gelatin-including bioink solution, the effect of printing temperature was expected to be more dramatic. Indeed, rheological characterization of this bioink solution at 25 °C exhibited higher viscosity throughout the relevant shear rate spectra (0.1–10 s^−1^) and higher G′/G″ ratio at low frequencies, compared with the same bioink solution at 28 °C (Figure 2, middle). In terms of applicability, more viscous bioink solution would require application of higher pressure through the nozzle to extrude the bioink, which in turn could be harmful for the cells. Thus, it was decided to work with a printing temperature setting of 28 °C when using this bioink solution.

To obtain even more increased mechanical stability of the core construct, Matrigel was added to the bioink solution (Table 1, bioink 3). Matrigel is commonly used for cell delivery in different cell-based strategies [48,49,50], therefore its inclusion could benefit residing cells during and post printing. Unlike gelatin, Matrigel gelation temperature is in the range of 22–37 °C [51]. To compensate for this effect, gelatin content in the bioink solution was reduced, resulting in a less viscous bioink (Figure 2B, right) compared with bioink containing higher gelatin concentration (Figure 2B, middle). However, the obtained bioink solution also exhibited gel-like behavior at low frequencies (Figure 2A, right), similar to higher gelatin content bioink solution. The mechanical stability of the Matrigel-containing bioink solution, and its rheological attributes were also examined at 37 °C. At this temperature setpoint, the mechanical spectra of this bioink increased dramatically compared with the 26 °C setpoint (two-fold increase at frequencies between 0.1–1 Hz, Figure 2A, right), implying the addition of Matrigel may contribute to the fabricated construct once incubated in culture.

#### 2.2.2. Core 3D Bio-Printing Optimization

To achieve the major advantage of 3D bio-printing, which is control of specific localization of desired components within the printed constructs, the optimal printing properties must be found. Depending on the bioink mechanical properties, three main factors affect the resulting 3D construct and its biological components: (a) biomaterial temperature; (b) applied pressure (to extrude the bioink from the needle) and printing velocity. Since each bioink has different properties, optimization for every bioink solution was performed independently, starting with the cell-laden inner core compartment.

In general, selected evaluation criteria for bio-printing quality was printing resolution, meaning the accuracy of printed strands placement, and strand thickness printed compared with desired strand thickness, depending on the type of needle used [52]. In the presented system, 250 µm needles were chosen for printing to achieve strands with respective thickness, optimal for cell viability (in terms of oxygen diffusion and nutrient supply), desired accuracy (resolution is dependent on needle size) and printed construct stability. Needle diameter also affects the printing pressure, as *p* ≤ 0.4 bar resulted in needle clogging, dictating a working range of 0.5–1 bar.

For cell visualization within the construct, GFP-expressing human fibroblasts (HF-GFP) were mixed with the polymer-based bioink solutions (detailed in Table 1), to achieve cell concentration of 10 × 10^6^ cells/mL, as will be required for subsequent fabrication of the CP. Selected geometry was 7 mm in diameter, 2 mm high cylinder (Figure 3A). Printing parameters were determined for each bioink solution, relating to the rheological analysis (for temperature settings) and printing capability; final printing parameters are detailed in Table 2. To evaluate core bioink printability, confocal imaging of printed constructs was used to reconstruct the fabricated 3D structures. When 3D bio-printing was accurate, organized grid patterns were clearly observed, as cells were located within the grid (Figure 3B).

The second criteria the bioink solution must answer for successful printing of the inner core of the patch is providing the cells with suitable biochemical cues allowing cell–matrix interactions, mimicking the native ECM, allowing the cells to organize in 3D. To evaluate the ability of cells to form 3D structures within the 3D printed constructs, the cell cytoskeleton was stained and visualized by confocal microscopy, estimating cell aspect ratio by image analysis (Figure 3C,D).

The 1.7% Alg-RGD, 0.36% Ca^2+^ bioink fabricated structures, although containing the highest content of RGD peptide among tested bioinks, presented inferior 3D morphology following 6 days in culture, compared with cells printed within 1.5% Alg-RGD, 0.3% Ca^2+^ and 3% gelatin bioink (Figure 3C, left and middle). Quantification of 3D cell morphology confirmed this observation (Figure 3D). The initial aspect ratio presented in the former constructs was significantly higher but did not change after 6 days (median value of 1.28). On the other hand, gelatin-containing bioink presented a significant increase in aspect ratio over time, with a median value of 1.18 on day 1 and 1.38 after 6 days. To exclude the theory that gelatin residues are responsible for the improved 3D morphology, the 1.5% Alg-RGD, 0.3% Ca^2+^ and 3% gelatin formulation was compared with the alternative of pristine, unmodified alginate hydrogel. Cells presented circular morphology in 3D in the absence of RGD modification (Appendix A
Figure A1). When Matrigel was added to the bioink formulation (bioink 3), cells exhibited improved capability to organize in 3D on day 1 compared with the 1.5% Alg-RGD, 0.3% Ca^2+^, 3% gelatin formulation (median value of 1.41 on day 1). However, median values of aspect ratio were similar by day 6 (1.45).

Since 3D bio-printing also affects cellular viability, the metabolic activity within the printed constructs was measured by a viability assay over 7 days in culture (Figure 3E). For both bioink formulations tested, cellular constructs maintained their metabolic rate with no apparent compensation in cellular viability observed. Overall, it was decided to continue with the formulation of 1.5% Alg-RGD, 0.3% Ca^2+^, 3% gelatin for cardiac patch fabrication, which exhibited the most promising potential to contribute 3D cell organization and survival.

#### 2.2.3. Shell Bioink Solutions

The shell bioink solution consists of three main components: calcium crosslinked alginate, the encapsulating matrix; alginate sulfate (AlgS), which is very low MW alginate polymer, modified with sulfate groups and capable of affinity-binding interactions with heparin-binding domains (discussed in 2.4); and EVs, the bioactive component, intended for delivery from the outer shell. For the shell bioink solution, three parameters were evaluated for their effect on the viscoelastic properties of the bioink: (a) presence of added components to encapsulating alginate matrix (Figure 4A); (b) working temperature (Figure 4B); and (c) AlgS content (Figure 4C). All shell bioink formulations included 1.5% final concentration of alginate (Alg, *w*/*v*), with final concentration 0.36% calcium gluconate (*w*/*v*). For examination of each component, the bioink either contained 0.018% AlgS (*w*/*v*) with 1 × 10^11^ EVs/mL, AlgS without EVs or no additives.

The addition of AlgS reduced the solution viscosity at low shear rates (0.1–10 s^−1^, Figure 4A(ii)) in the 1.5% Alg formulation. On the other hand, the addition of AlgS and EVs together slightly moderated this reduction. However, each component addition influenced the viscoelastic response of the bioink, indicated by the location of the cross point between storage modulus G′ (elastic response) and the loss modulus G″ (viscous response). At low angular frequencies, G′ exceed G″, meaning the solution is at solid-like phase. With increased frequency, a cross point appears, indicating phase transition known to occur with such partially crosslinked solutions. Since neither AlgS nor EVs contribute to crosslinking degree (do not participate and interrupt the electrostatic interactions between alginate polymer chains and calcium ions), phase transition occurs at lower frequencies (Figure 4A(i). These results mean that 1.5% Alg, 0.36% Ca^2+^ supplemented with AlgS and EVs could be injected at relatively low shear rates. With additional cross-linking (using CaCl_2_), this hydrogel will solidify and could form a stable printed construct.

Since the presented bioink formulation exhibits liquid-like viscoelastic properties, the next step was to explore the effect of temperature (Figure 4B), examining two setpoints: 16 (blue) and 18 °C (red). Even though the viscosity spectrum was barely altered by the change in temperature (Figure 4B(ii)), the solution behavior changed dramatically at 18 °C, exhibiting liquid-like properties at very low frequencies (Figure 4B(i)). Decreased bioink temperature resulted with a shift in the solution cross point, indicating the solution is capable of phase transition at that temperature.

For AlgS content effect evaluation, both 18 and 16 °C were examined, keeping final unmodified alginate and calcium concentration constant (Figure 4C). Interestingly, increasing AlgS concentration affected mostly the viscosity of the bioink solution, presenting higher viscosities at both examined temperatures compared with lower AlgS content (Figure 4C(i)). This was also evident by the ratio between G′ and G″ range (Figure 4C(i)). Nevertheless, no change in the cross-point location was observed because of AlgS increased amount.

Following characterization of the shell bioink mechanical properties, 3D bio-printing of these solutions was examined to determine the appropriate printing setting best suitable for fabrication of accurate macrostructures. The outer shell is designated for sustained delivery of EVs, therefore should be printed at a high enough resolution to allow control of shell porosity, affecting the erosion rate of the hydrogel, mechanical stiffness of the construct (since it surrounds the entire patch) and mass transfer into the core compartment. To examine printing accuracy, 0.5 mm thick, grid patterned constructs were printed and strand thickness was measured (Figure 4D). Considering the liquid-like nature of shell bioink solution at low frequencies and shear rates, it was decided to examine the effect of the two parameters that altered solution behavior, printing temperature and AlgS content. Since printing pressure also dramatically changes the amount of material applied during printing, it was decided to focus the printing optimization efforts at finding the appropriate printing velocity, allowing more fine tuning than the pressure parameter. Printing pressure was set at 0.3 bar, as higher pressure resulted in uncontrolled disperse of the bioink while printing, while lower printing pressure led to nozzle blockage.

In general, printing of solution with 0.018% AlgS exhibited the most reproducible results, evident from the strand thickness distribution obtained at both examined temperatures (Figure 4D, bottom left and middle). For bio-printing of 0.045% AlgS, only the 18 °C setting produced sustainable constructs (Figure 4D, top right), however presented high variability in strand size distribution, forcing application of high printing velocity of 22 mm/s to produce accurately printed strands (Figure 4D, bottom right). Even though these settings resulted with high printability (0.983 ± 0.012), they also led to higher variability compared with the best settings for shell bioink solution containing 0.018% AlgS (16 °C, 0.3 bar and 18 mm/s; 0.981 ± 0.008). Therefore, the latter conditions were chosen for 3D bio-printing hereafter.

### 2.3. Two-Component Cardiac Patch Fabrication

After characterization of the bioink solution properties and setting the suitable printing parameters, we moved on to fabrication of the two-compartment CP, according to the proposed design (Figure 1). To this end, the 3D bioprinter was installed with two printing syringes: one for the shell, loaded with 1.5% Alg, 0.36% Ca^2+^, 0.018% AlgS (all *w*/*v*) bioink solution and the other for the cell-laden core, loaded with any of the tested core bioink solutions, discussed in Section 3.2. As in most 3D bio-printing methodologies, constructs were fabricated layer-by-layer, assigning each bioink to the designated compartment, printed simultaneously.

For proper visualization of printing outcome, fluorescently labeled components were used in each bioink: AlgS for the shell, and GFP-expressing cells for the core (Figure 5A(i)). Confocal microscopy was used for 3D reconstruction of the fabricated construct, allowing imaging of each layer separately, confirming successful implementation of the two-compartment design (Figure 5A(ii)). Confocal imaging also validated the high printing resolution of the shell bioink, as the 400 µm thick, grid pattern is clearly visible at the bottom.

Young’s moduli measurements of the 3D bio-printed CP emphasized the major contribution each component in the core bioink solution has on resulting stiffness in general, but also on the changing kinetics over a span of 5 days (Figure 5B). The 1.7% Alg-RGD, 0.36% Ca^2+^ fabricated cores resulted in the least sustainable constructs to begin with (1.1 ± 0.2 kPa), often significantly degraded following 7 days in culture (unless delicately handled). The addition of gelatin to the bioink formulation resulted in CP with initial superior mechanical stiffness (4.2 ± 0.4 kPa), gradually decreasing with time (2.0 ± 0.1 kPa after day 5). Unlike the former bioink, these constructs were easier to handle and maintained integrity for more than 2-weeks in culture. Bioink solution formulation that included Matrigel and contained lower gelatin content results with the most stable CP. Furthermore, addition of Matrigel also changed the stability kinetics of the 3D bio-printed constructs, presenting three-fold increase in stiffness on day 2 relative to day 1 (15.4 ± 0.3 kPa compared with 5.6 ± 0.7 kPa, respectively), instead of gradual decrease or maintenance with time observed in the former two bioink formulations. Following 5 days in culture, the construct’s elastic modulus was decreased, however was still significantly higher compared with its initial stiffness (9.5 ± 0.7 kPa). Since only the two bioink formulations that included gelatin provided sufficient initial mechanical stability, it was decided to continue only with those bioink solutions for inner core fabrication.

### 2.4. EVs Interacts with Alginate Sulfate through Affinity-Binding

The uptake of EVs by recipient cells is rapid. Hence, a controlled, sustained release system is required to prolong EV therapeutic effect. It was previously demonstrated that EVs could be isolated from a solution using heparin-coated beads [53]. To this end, we aimed at adjusting the well-established release matrix, where alginate is the encapsulating matrix and alginate sulfate is the component affecting sustained release through affinity binding of the EV’s heparin binding domains [31,32,33,54].

First, the idea of loading and releasing of EVs via affinity binding to AlgS must be applicable. Thus, the dissociation constant of EVs to AlgS was compared to pristine alginate and heparin, using Surface Plasmon Resonance (SPR) spectroscopy. All three ligands were covalently bound to a sensor chip, as EVs in different concentrations were used to determine the dissociation constant (Figure 6). As expected, pristine alginate did not present any response to EV flow (Figure 6A), showing EVs do not interact with unmodified alginate. However, both AlgS and heparin exhibited a similar binding curve, typical for affinity binding (Figure 6B and C, respectively). From the plateau region of the binding sensograms, K_D_ values were extracted and estimated to be 3.26 ± 0.51 × 10^−15^ M for AlgS and 7.62 ± 1.42 × 10^−15^ M for heparin. These findings suggest that alginate sulfate can bind EVs through affinity binding, and therefore it has the potential to affect EV release when encapsulated with a polymeric matrix.

### 2.5. Three-Dimensional Bio-Printing of AlgS-EVs Complexes Encapsulated within Alginate Hydrogel

Since AlgS and EVs are part of the delivery component of the proposed CP design, the next goal was to locate EV-AlgS complexes throughout the printed construct’s shell. Application of EV affinity binding was identical to protein affinity binding, as previously described [31,32,34]. EVs were incubated with AlgS polymer for 1.5 h at 37 °C, then mixed with crosslinked alginate matrix to form the bioink (Figure 7A), later 3D bio-printed according to previously determined printing parameters.

To validate EV presence inside the 3D bio-printed matrix, two methods were applied: cryo-scanning electron microscopy (Cryo-SEM Figure 7B) and confocal microscopy (Figure 7C), both provide the required high resolution for nano-scale molecules or aggregate imaging.

Cryo-SEM analysis of the bio-printed construct fabricated by bioink including AlgS-EV complexes or with AlgS without EVs (Figure 7B(i) and (ii), respectively) enabled to distinguish EVs within the matrix. Application of Cryo-SEM imaging also allowed preservation of EV structure (since samples are not dehydrated), showing EVs maintain sphere-like morphology (Figure 7B(iii)). Energy dispersive spectroscopy (EDS) analysis of EV presenting regions indicated presence of organic matter and phosphorous, indicating these are indeed membrane enclosed nano-vesicles (Figure 7B(iv)).

Next, the same bioink was used for 3D bio-printing of the outer shell compartment of the CP. Printed constructs were examined by confocal microscopy, using fluorescently labeled EVs and AlgS for outer shell compartment visualization (Figure 7C,D). To distinguish between printed layers, volume projections were reconstructed by capturing the planes ranged from slides bottom up to 1.5 mm deep (Z-stack steps were 150 µm; Figure 7C,D). According to the CP design, AlgS-EVs complexes should be present throughout the bottom half of the construct (0.4 mm high) and at the upper ring (1.2 mm thick). As expected, the bioactive complexes were printed along the grid pattern throughout the bottom of the CP (Figure 7D, middle). At higher layers, starting 0.4 mm from patch base, only a ring-shape of fluorescent complexes was observed (Figure 7D, right), confirming successful localization of EV-AlgS complexes in the designated places.

### 2.6. Alginate Sulfate Prolongs EV Release Profile from 3D-Printed Alginate Matrix

The next step was to determine release kinetics of affinity bound EVs from alginate matrices. First, release of EVs from the matrix was evaluated in a simplified model of alginate hydrogel beads (1.35% Alg, cross-linked in CaCl_2_ bath), incorporated with either affinity-bound or simply encapsulated EVs (Figure 8A). EV cumulative release was quantified by Nanoparticle Tracking Analysis (NTA). Surprisingly, it seems that EVs encapsulated with Alg-AlgS polymer combination are more readily released from the alginate beads (the cumulative effect was statistically significant at day 5), as opposed to the initial assumption that EVs will be released slower from such matrix. Moreover, all alginate beads were dissociated by day 7 (both Alg and AlgS matrices), implying the model of beads reflects more on the stability of the beads rather than the effect of affinity binding. Also, worth noting is that the variability between samples exhibited in this model was high in both groups, making it more difficult to interpret. One explanation for this observation is related to alginate bead preparation. Since alginate beads were crosslinked in a CaCl_2_ bath, these beads exhibit surface erosion-dependent release mechanism. AlgS does not crosslink with bivalent ions, and therefore interferes with crosslinking of alginate chains within the beads, resulting in a mechanically inferior hydrogel in its core. Hence, this model emphasized the pivotal role of the calcium crosslinking method on the obtained release profile from alginate, especially when the hydrogel contains AlgS.

Nevertheless, the workflow of 3D bio-printing of the CP presents different methodology, as the bioink is made by first partially crosslinking the alginate matrix, then adding the AlgS-EV complexes, and finally printed into calcium containing FRESH, then further crosslinked with 0.1 M CaCl_2_. Since the delivery system is intended to be incorporated within the CP outer shell, the release kinetics were also evaluated using the 3D bio-printed patch model (Figure 8B). To examine the effect of alginate sulfate on EV release from the CP, outer shell bioink was prepared with either affinity bound or encapsulated EVs. After printing, the constructs were incubated in DMEM medium until no EVs were observed in incubation media.

In general, EV release kinetics from alginate constructs seem to be rapid. (Figure 8B). Over 75% of total EVs content was released within the first 48 h of incubation. After 7 days, more than 90% of EVs were already released. As for the effect of AlgS on EV release, the relative amount released did alter significantly by day 4 when EVs-AlgS complexes were used (87.7 ± 1.2% compared with 92.0 ± 0.4% without AlgS). Moreover, release of EVs from AlgS-containing constructs was prolonged by two days compared with 0% AlgS constructs (media contained less than the detection threshold after 11 days). To summarize, AlgS affinity binding has a slight effect on release of EVs over time, sustaining its release from the 3D bio-printed matrix.

## 3. Discussion

### 3.1. Cardiac Patch Design and 3D Bio-Printing of Inner Core

The main aim of this work was to design an alginate-based scaffold, for fabrication of a cell-laden CP combined with EV delivery system. To accomplish each criterion, the CP design included two distinct compartments, one including the cellular components and ECM mimicking features; and the other designated for EV controlled release (Figure 1A). The proposed design features the delivery compartment at the outer shell of the patch, surrounding the cell-laden core symmetrically (400 µm thick on each side and perimeter) to avoid orientation concerns upon transplantation. To fabricate such a 3D construct, with distinct and accurate spatial placement of each component, a 3D bio-printing approach was implemented, using alginate-based bioink solutions. According to the compartment role, the bioink solutions had to answer the requirements regarding mechanical properties, biomaterial macrostructure and biochemical microenvironment.

In the case of both core and shell bioink solutions, the primary concern was their printability, dictated by the viscoelastic characteristics of the solution. Here, the mechanical properties of partially crosslinked alginate-based solutions were examined (Figure 2 and Figure 4).

For the cell-laden core compartment, three bioink solution formulations were examined, all were based on RGD-modified alginate (Alg-RGD), differed by the addition of different polymers (final compositions detailed in Table 1). Alginate has been widely investigated as an injectable hydrogel for cell delivery, mostly used in low concentrations for these purposes [55,56]. For 3D bio-printing applications, the use of alginate requires printing using FRESH, providing temporary physical support, accompanied with additional ionic crosslinking to stabilize the 3D construct. Partially crosslinked Alg-RGD also presents additional unique mechanical properties—the storage (G′) and loss (G″) moduli of the solution are closely related or share a cross point. This physical behavior indicated the crosslinked material is capable of phase transition from its liquid state into a hydrogel [57]. Such behavior was observed in the bioink solution containing only Alg-RGD (bioink 1), exhibiting a G′/G″ ratio close to 1 almost throughout the entire angular frequency spectra (2A, top). When gelatin was added to the solution, both G′/G″ ratio and solution viscosity increased dramatically, exhibiting temperature dependent behavior (bioink 2, Figure 2A,B, middle). When gelatin content was decreased (compensated with the addition of Matrigel to the bioink), the resulting bioink solution exhibited a moderate decrease in both parameters, yet higher than those observed without gelatin at all. These findings correlate with those of others, indicating that the addition of gelatin to the bioink increases its viscosity [58].

However, the addition of gelatin in the bioink solution also sets more restrictions regarding applicable printing parameters. Gelatin mechanical properties are temperature dependent, therefore enforcing a narrow temperature working range, between 24–30 °C [59]. Since gelatin addition also increases the viscosity of the bioink solution, a higher inlet pressure is required. For cell-laden applications, the applied pressure is restricted to 1 bar, otherwise risking a decrease in cellular viability [60,61]. In light of these restrictions, the printing parameters for each bioink solution were set accordingly (Table 2), producing acceptable printing, indicated by obtained printed strands (Figure 3B) and cellular viability within 3D bio-printed constructs. The effect of bioink composition on cellular’ 3D organization was also examined in a human fibroblast model. The 3D constructs printed using gelatin containing bioink solution exhibited a higher aspect ratio, indicating encapsulated cells could form cell–matrix interaction better in these constructs (Figure 3C,D). Since gelatin consists of RGD residues as well, it was speculated that the observed increase in cell 3D organization is due to gelatin residuals remaining even after gelatin solubilization. This explanation was ruled out once pristine alginate-gelatin composites exhibited circular 3D morphology (Figure A1). Even though all bioink solutions include alginate modified with RGD peptides, required for 3D organization [62,63], one should keep in mind that when cell-laden bioink solutions are used, the cells are encapsulated within the matrix. Therefore, 3D organization over time also depends on the matrix degradation or erosion kinetics, affecting its porosity. This could be the case in the 3D bio-printed constructs composed of Alg-RGD–gelatin composites. Another indication supporting this claim is the 3D morphology observed in constructs printed with Alg-RGD–gelatin–Matrigel composites, presenting constant aspect ratio over time. This is possibly due to Matrigel gelation process occurring during incubation of the construct, limiting free space for cells to spread and form interactions with the matrix.

One of the key attributes for successful implementation of a CP is its mechanical stability. In fabrication of cell-laden 3D structures, and particularly patches, there is a tradeoff between the whole construct stiffness and inner structure elasticity and porosity. While the former is important for patch stability, ultimately influencing patch retention post transplantation, the latter affects the cellular organization and therefore functionality. Therefore, the effect of different core bioink solutions used for fabrication of the inner core on the patch mechanical stiffness was studied in parallel to the bioink’s influence on the residing cells. Indeed, the polymer composition of the inner core affected the value of elastic moduli of the CP, exhibiting different dynamics over time (Figure 5C). The inclusion of gelatin (4 and 3 kPa by days 1 and 2, respectively) and Matrigel (5 and 15 kPa by days 1 and 2, respectively) provided sufficient initial stiffness to the CP. This attribute is important for patch sustainability upon transplantation, compelling elastic modulus between 0.1–20 kPa [64]. Differing from the composite bioink solutions, CPs consisting of only Alg-RGD presented inferior stiffness (below 1 kPa throughout 5 days in culture), dissociating over time and exhibiting poor sustainability. Altogether, the gelatin-containing bioink solutions exhibited the best results with regards to cellular viability and 3D behavior within the CP in a human fibroblasts model, along with the required stiffness for cardiac implementation.

### 3.2. Three-Dimensional Bio-Printing of Outer Shell and Alginate Sulfate—EVs Delivery System

Contrary to the core bioink solutions, the shell bioink printing parameters, namely the temperature and inlet pressure, were less strict to begin with. EVs were shown to keep their membrane stability even when subjected to relatively high shear stress (>1 bar) [65,66]. Nevertheless, the design of the delivery compartment demanded higher bio-printing resolution to allow fabrication of controlled porosity of the printed pattern, allowing proper mass transfer to the CP interior. Similar to characterization of the core bioink solutions, the rheological testing of the shell bioink provided important insights into the parameters affecting printability (Figure 4A–C). As discussed above, partially crosslinked alginate is capable of phase transition, making it suitable for 3D bio-printing. The shell bioink exhibited a temperature dependent mechanical behavior which is also in line with previous studies [67]. Interestingly, AlgS content did not affect cross point location. These observations seem odd, since AlgS is not capable of crosslinking with calcium due to the polymer sulfation, disrupting the formation of crosslinking ion bridges [23]; therefore, it is expected that increased AlgS content would result with more liquid-like behavior compared with the bioink solution. However, in both bioink solutions, AlgS constitutes less than 5% of total alginate content. It is possible that the gel-like behavior observed in low frequencies is a result of AlgS aggregation into NPs (occurring when AlgS is highly concentrated, during bioink preparation). This explanation seems to correspond with the effect of EV addition to the bioink solution, slightly increasing its viscosity. According to the mechanical behavior of the shell bioink, printability analysis of this bioink was performed (Figure 4D), indicating that the 1.5% Alg, 0.36% Ca^2+^, 0.018% AlgS bioink formulation, printed at 16 °C, 0.3 bar at 18 mm/s printing velocity produced the most accurate grid pattern (0.981 ± 0.008), exhibiting low variability of printed strands (Figure 4D(ii)).

AlgS plays a key role in the presented delivery system. It was previously demonstrated to interact with a variety of growth factors and cytokines through affinity binding interactions, enabling prolonged and sustained release from alginate scaffolds [31,34] and injectable hydrogels [26,32,33]. It was also shown that EVs present heparin-binding domains on their membrane, allowing them to interact with Heparin through affinity binding [53]. Therefore, it was speculated that EVs could also interact with AlgS, similarly to Heparin. This was confirmed by SPR analysis, showing that the dissociation constants (K_D_) of EVs with AlgS and Heparin had similar values (Figure 6). This finding sets the ground for an additional mechanism of sustained release of EVs from polymer-based matrices, mediated by affinity binding. Thus far, EVs sustained delivery was only attempted by encapsulation inside a scaffold or hydrogel, making matrix degradation the limiting factor for drug release rate [66,67,68].

According to the proposed design, the two-compartment CP was successfully fabricated using the core and shell bioink solution, printed simultaneously (Figure 5A and Figure 7C). Cryo-SEM imaging validated that the 3D bio-printed EVs maintained their morphology (Figure 7B), while confocal microscopy was used to confirm that the AlgS-EVs complexes were indeed located successfully in the patch shell (Figure 7C,D). To date, EV-based delivery systems have been usually fabricated by encapsulation of EVs within the entire hydrogel construct. Even though EVs have been used as bioink for 3D bio-printing (using glycerol [65]), incorporation of EVs within biomaterial-based hydrogels in designated locations has not yet been attempted. The capability to concentrate the EVs within a specific location has consequences on the efficiency of this delivery system. Homogeneous encapsulation of the EVs requires larger amounts of EVs, while smart distribution may be less squandering and even beneficial when only a local effect is required.

The release profile of EVs from the 3D bio-printed CP was shown to be affected by pre-incubation of AlgS with the EVs prior to addition to the bioink, displaying a prolonged release profile of EVs from these constructs (Figure 8B). Still, the release profile from the alginate matrix was considered rapid, as more than 75% of the EVs were released within 2 days. However, other EV delivery systems also exhibited an initial burst release with similar cumulative release profiles [66,69]. The obtained release profile, along with the mechanical stiffness of the CP (Figure 5B), dictate that the optimal time point for future applications involving patch implementation will be at most 24 h after printing.

### 3.3. Research Outlook and Future Prespective

Individually, utilization of 3D bio-printing for cell-laden scaffold fabrication (i.e., for construction of bone [70], vasculature constructs [71,72] and CPs [73,74]) or for drug loading and delivery purposes [75,76,77] has been vastly investigated in recent years. Nevertheless, the concept of combining these strategies into a single, dual-functional structure was never attempted. In cardiac tissue regeneration (but not exclusively [78]), the notion that the cellular secretome is the main driving force behind tissue regeneration has been suggested [4,79,80]. Therefore, combining the cellular component with a supporting, drug delivery platform of bioactive molecules inducing regenerative processes, introduced in this research, is of extreme importance. Furthermore, by localizing the delivery platform at the designated patch-host interface, this design has the potential to better introduce these bioactive molecules to their target, thereby affecting treatment efficacy.

However, since patch integration needs to be evaluated in a cardiac injury model, the effect of EV delivery in vivo remains ambiguous. One possible future direction includes incorporation of EVs, specifically designed to induce cardiac regeneration related processes (e.g., attenuation of CM death, CM proliferation and promotion of angiogenesis) within the CP shell. Such engineered EVs could induce cardiac regeneration directly at the interface, hypothesized to contribute to CP assimilation following cardiac injury and thus improving the efficacy of treatment.

## 4. Conclusions

Herein, a bi-functional CP design was introduced, incorporating a cell-laden core with an EV-laden compartment at its shell for better integration of the CP with the host tissue. The proposed design was successfully fabricated using extrusion-based 3D bio-printing following a comprehensive study for optimizing the printing conditions for each bioink solution composing the shell and core compartments. Incorporation of AlgS in the outer shell compartment enabled the binding of EVs via affinity binding interactions. The EVs were shown to maintain their integrity following 3D bio-printing. The resulted 3D-bioprinted dual-compartment CP exhibited elastic modulus values relevant for cardiac tissue development and for future transplantation. Furthermore, the CP demonstrated sustained release of EVs. To the best of our knowledge, and even though EVs were recently incorporated within injectable hydrogels and bioink solutions [81,82,83], the concept of combining an EV delivery system within a cell-laden construct was never attempted. This approach holds great potential for promoting more efficient patch assimilation with the host, containing the tissue substitute component necessary for function restoration, while applying spatial, concentrated EV-based cell signaling at the interface. Incorporation of functional EVs, either natural [84,85] or modified [86,87] within the proposed design and examination in a cardiac injury animal model is required to further evaluate the efficacy of this approach.

## 5. Materials and Methods

### 5.1. Materials

Roswell Park Memorial Institute (RPMI) 1640 medium was from Gibco (Gaithersburg, MD, USA). Dulbecco’s Modified Eagle medium (DMEM), Sodium Pyruvate, L-glutamine, penicillin/streptomycin, heat inactivated fetal bovine serum (FBS) were from Biological Industries (Kibbutz Beit-Haemek, Israel). Other reagents, unless specified otherwise, were purchased from Sigma-Merck (Rechovot, Israel). Sodium alginates (VLVG, 32 kDa and LVG; 100 kDa, >65% guluronic acid monomer content) were from FMC Biopolymers (Drammen, Norway). Alginate-sulfate (AlgS) was synthesized from VLVG as previously described [31]. All reagents were of analytical grade.

### 5.2. Cell Culture

Human monocytic THP-1 cells (ATCC^®^ TIB-202, VA, USA) were cultured in RPMI 1640 supplemented with 2 mM L-glutamine, 1 mM sodium pyruvate, penicillin (100 U/mL) and streptomycin (100 µg/mL). THP-1 monocytes were seeded at a density of 0.3 × 10^6^ cells/mL and differentiated into macrophages (MΦs) by incubation with 100 ng/mL phorbol 12-myristate 13-acetate (PMA) in serum-free RPMI medium for 2 days followed by 24 h incubation in RPMI culture medium. MΦs activation was performed by 24 h incubation with serum-free RPMI medium, supplemented with 20 ng/mL human recombinant IFN-γ (Peprotech^®^, #300-02) and 10 pg/mL of LPS.

Foreskin human fibroblast cells expressing GFP (HF-ARZ, GFP-HF) were seeded at a density of 4 × 10^4^ cells cm^−2^ and grown to confluence in DMEM, supplemented with 10% (*v*/*v*) FBS, 100 U/mL penicillin, 0.1 mg/mL streptomycin and 1% L-glutamine (*v*/*v*)).

### 5.3. Extracellular Vesicles Isolation and Characterization

#### 5.3.1. Extracellular Vesicles Isolation

EVs were isolated from culture media using differential centrifugation and ultrafiltration methodologies. Activated THP-1 derived MΦs were washed with PBS and replaced to fresh, 22 mL of serum-free RPMI 1640 medium. To increase EV yield, the medium was supplemented with 100 mM ethanol, previously shown to elevate EV secretion levels [88,89]. After 48 h incubation, culture media was collected on ice, centrifuged at 3500 g for 10 min at RT, to remove cell debris. Supernatant was collected and centrifuged at 10,000 g for 30 min for apoptotic bodies elimination. Supernatant was filtered through a 0.22-µm filter to eliminate larger vesicles, followed by two-step ultrafiltration, using 100 kDa MWCO Amicon™ centrifugal filters (Sigma-Merck), according to manufacturer’s protocol.

#### 5.3.2. Nanoparticle Tracking Analysis (NTA) of Size and Concentration

Size distribution and concentration of isolated EVs were measured using NanoSight NS300 system (Malvern, UK). EV samples were diluted in PBS until individual nanoparticles could be tracked. The samples were captured for 60 s at room temperature (RT). NTA software was used to measure particle concentration (particles/mL) and size distribution (in nanometers). For each sample, five measurements were taken, and the mean value was determined.

### 5.4. Surface Plasmon Resonance (SPR) Analysis of the Molecular Interactions of Alginate Sulfate with EVs

Real-time biomolecular interaction studies were performed using the BIAcore 3000 instrument (Pharmacia, Uppsala, Sweden), operated using BIA evaluation version 3.2 Software. Studies were performed as previously described [31,34]. Briefly, SA sensor chip priming was performed prior to immobilization of biotinylated polysaccharides using sequential pulses of 70% (*v*/*v*) glycerol, followed by pulses of 50 mM NaOH and 1 M NaCl to remove residues of non-covalently bound streptavidin from the sensor chip. All four channels of the SA sensor chip were used, maintaining the first flow cell #1 (FC-1) without polysaccharide immobilization for reduction of non-specific binding purposes. Three samples of Biotinylated polysaccharides of heparin, serving as a positive control, alginate-sulfate and pristine alginate were immobilized onto flow cells 2–4, respectively. Binding measurements were performed over a range of concentrations of EVs. The EVs were diluted with PBS buffer immediately prior to injection over the channel (flow rate of 20 mL/min, 4 min, followed by 3 min dissociation time). Data points were acquired continuously throughout the binding and dissociation phases, and analysis was based on the response once reaching saturation binding. The real-time curve derived from the reference channel (FC-1) was subtracted from the binding curves obtained from the immobilized polysaccharide-containing flow channels. Association and dissociation rate constants were calculated by nonlinear curve fitting of the primary sensorgram data using the Langmuir binding model available in the BIA evaluation 3.2 Software. From the plateau region of the binding sensogram, the dissociation rate constants (K_D_) were extracted, using BIA evaluation version 3.2 Software.

### 5.5. Three-Dimensional Bio-Printing

#### 5.5.1. Shell Bioink Solution Preparation

All shell bioink solutions were based on partially crosslinked alginate solutions. Calcium crosslinked alginate solutions were prepared by mixing 4 mL of 2.5% *w*/*v* alginate with 1 mL of 2.4% *w*/*v* D-gluconic acid salt solution, using homogenization to distribute the calcium ions throughout the solution, further stirred at RT until used. Next, 2.4 mL of the crosslinked alginate solution was loaded into a 3-mL syringe, then connected to an additional 3-mL syringe through a Luer-to-Luer connector. The second syringe was loaded with 0.8 mL of either DDW, alginate sulfate (AlgS, in desired concentration) or AlgS-EVs mixture. The solutions were mixed by gently pushing the pistons back and forth for 1 min, until homogenously mixed.

#### 5.5.2. Core Bioink Solution Preparation

Three core bioink solutions were examined, all containing partially crosslinked RGD-modified alginate solution (Alg-RGD). Final compositions of the used cell-laden bioink solutions are detailed in Table 1. Alginate was covalently modified with RGD peptide as previously described [90]. Alg-RGD solution was prepared by dissolving lyophilized alginate-RGD in DDW and allowed to dissolve under stirring for 1–2 h. Calcium crosslinked alginate was prepared by mixing 2.5% *w*/*v* Alg-RGD with of 3% *w*/*v* D-gluconic acid salt solution (6:1 volumetric ratio), using homogenization to distribute the calcium ions throughout the solution, further stirred at 37 °C until used. 15% gelatin stock solution was prepared by dissolving bovine gelatin Type B in DMEM medium and allowed to dissolve under stirring for 1 h at 37 °C. When gelatin-containing bioink solutions were used, gelatin stock solution was added to the crosslinked alginate, allowed to mix for 10 min. Before cell addition, the polymer mixture was transferred to a sterile syringe. GFP-expressing human fibroblasts (HF-GFP) were loaded into an additional sterile syringe. In case Matrigel was added to the bioink, cell pellets were first resuspended in Matrigel at 4 °C, prior to mixing. The two syringes were immediately connected through a Luer-to-Luer connector. The solutions were mixed by gently pushing the pistons back and forth for 1 min, until homogenously mixed. The final mixture consisted of 1 × 10^7^ cells/mL.

#### 5.5.3. Rheological Characterization of Bioink Solutions

The viscoelastic properties of core and shell bioink solutions were analyzed using a stress control Rheometer (TA Instruments, model AR 2000), operated in the cone-plate mode with a cone angle of 1° and a 60 mm diameter. Storage (G′) and loss (G″) moduli were measured in a frequency range of 0.1–10 Hz, while the apparent viscosities (Pa*s) of the bioink solutions were assessed at different shear rates between 0.1 to 10 s^−1^. The measuring device was equipped with a temperature control unit (Peltier plate, ±0.05 °C) operated at desired temperature between 16–37 °C.

#### 5.5.4. Three-Dimensional Bio-Printing Procedure and Constructs Maintenance

Prior to 3D bio-printing, the bioink was deposited into sterilized 30-mL printer barrel sealed with a fit plunger, using a Luer-to-Luer connector. The printer barrel was then sealed and centrifuged for 1 min at 150 g to remove remaining air bubbles. A sterile 25-guage needle tip was added to the barrel, and a cap connecting the print head to the barrel was added. The barrel was put in the low-temperature head of the bioprinter (EnvisionTEC 3D-bioplotter Developer Series, Germany), set to appropriate temperature according to inserted bioink solution. The bioink was allowed to reach the printing head temperature for 30 min before start bioprinting.

Three-dimensional bio-printing was preformed using the Freeform Reversible embedding of Suspended Hydrogel (FRESH) approach, as previously described [91]. The 3D constructs were printed onto a 12-well plate platform filled with gelatin microspheres supporting bath at RT, using the printing parameters detailed in Table 2. For optimization of printing parameters, 3-layer constructs were printed using a single bioink, with an infill pattern of 90° grids with 0.8 mm spacing (center-to-center) and a 30% overlap between printed layers. For optimization of inner core studies, 13-layer constructs were printed using a single cell-laden bioink, with an infill pattern of 90° grids with 0.6 mm spacing (center-to-center) and a 30% overlap between printed layers. For cardiac patches (CP) fabrication, two types of bioink solutions were used simultaneously to print the 4-part construct: shell bioink solution for printing of CP outer shell (400 µm-thick base, middle ring and sealing cap), and cell-laden core bioink, used to print the interior compartment of the CP (1.2 mm thick, 9.2 mm in diameter). Overall dimensions of the CPs were 10 mm in diameter and 2 mm thick, which printed in 14 layers, with an infill pattern of 90° grids with 0.6 mm spacing (center-to-center) and a 30% overlap between printed layers. CPs were printed in triplicates. CAD models of the CP and grids printed were generated using SOLIDWORKS and imported to the printing control system through the Bioplotter RP software. Following bioprinting, CPs were incubated for 1 h at 37 °C, 5% CO_2_ to free the 3D printed constructs from the gelatin support bath. The CPs were then washed and incubated for 10 min at 37 °C in growth medium supplemented with 0.1 M CaCl_2_ to further crosslink the 3D bio-printed constructs. Following incubation, the CPs were incubated in growth medium, in new 12-well plate. Media was changed daily for each CP until analysis.

#### 5.5.5. Printability Analysis

The printability analysis examined the effectiveness of the extruded strands in the test grids to form square holes between strands, as previously described [92]. Circularity (C) of an enclosed area is based on the shape perimeter and area, where a perfect circle has a circularity of 1. For a square shape, circularity is equal to π/4. To this end, and as previously derived and defined, printability is given by Equation (1):(1)$$Pr=\frac{L^{2}}{16A}$$
where *L* is perimeter and *A* is area of a shape. *A* printability of 1 is equal to a perfect square and indicates optimal printing conditions of a bioink. Bright field images of test grids were evaluated by measuring the perimeter and area of several holes in each sample and printability was calculated, with 3–4 constructs and at least 20 holes were measured.

### 5.6. Cryogenic Scanning Electron Microscopy (Cryo-SEM)

The samples were placed and sandwiched between two aluminum discs (3 mm in diameter, 25 μm in thick each) and cryo-immobilized in a high-pressure freezing device (EM ICE, Leica).

The frozen samples were then mounted on a holder under liquid nitrogen in a specialized loading station (EM VCM, Leica) and transferred under cryogenic conditions (EM VCT500, Leica) to a sample preparation freeze fracture device (EM ACE900, Leica). In that device, the samples were fractured by a rapid stroke of a cryogenically cooled knife, exposing the inner part of the sandwiched discs. After fractured, the samples were etched at −100 °C for 10 min to sublime ice from the sample surface and coated with 3 nm carbon.

Samples were imaged in a Gemini SEM (Zeiss) by a secondary electron in-lens detector while maintaining an operating temperature of −120 °C. The measurements were done at the Ilse Katz Institute for Nanoscale Science and Technology Ben-Gurion University of the Negev, Beer Sheva, Israel.

### 5.7. Mechanical Stiffness of 3D Bio-Printed Cardiac Constructs

The elastic modulus (Young’s modulus) of 3D bio-printed CPs was analyzed by an Instron 4505 mechanical tester (courtesy of Prof. Ronit Bitton, Ben-Gurion University of The Negev, Israel) equipped with a 100 N load cell. The crosshead speed was set to 5 mm/min, and load was applied until the specimens were compressed to approximately 100% of the original thickness. The elastic modulus was calculated as the slope of the initial linear portion of the stress–strain curve (n = 3–5).

### 5.8. EVs Release Studies

#### 5.8.1. Preparation of Alginate/Alginate-Sulfate Hydrogel Microspheres with EVs

EVs (3 × 10^10^, dissolved in PBS) were added to either DDW or AlgS (in DDW, 50 µg; 1:50,000 molar ratio) and incubated for 1.5 h at 37 °C, to allow equilibrium binding. The EV-AlgS mixture was then mixed with 1.5% alginate solution (Alg, *w*/*v*) in a 1:9 volumetric ratio. The mixed solution (1.35% Alg, 0.0005% or 0% AlgS) was collected into a syringe with 18 G needle and dropped into stirred CaCl_2_ solution (10–12 mL, 0.15 M) to allow crosslinking of the alginate chains. The Ca-alginate hydrogel microspheres (~1 mm diameter) were allowed to stir at RT for 2 min. The microspheres were collected by filtration through 106 µm filter, then washed once with release media (DMEM without serum, 1% Pen-Strep). For release studies, 4 microspheres were suspended in 1 mL release medium and incubated on a rotating incubator, at 37 °C, 5% CO_2_. The medium was replaced every 2 days. To determine initial amount of EVs inside the microspheres, a sample of 4 microspheres was dissociated using 500 µL sodium citrate 4% (in PBS) to free encapsulated EVs. The sample was centrifuged for 10 min at 6000 g, 4 °C. EV-containing supernatant was collected, and volume was adjusted to 1 mL with PBS. The amount of EVs in the sample or released into the media was determined by NTA.

#### 5.8.2. Release from 3D Bio-Printed Constructs

Constructs were fabricated using 3D bioprinting as described in 5.4.4, using acellular 1.5% Alg-RGD, 0.3% Ca^2+^ and 3% gelatin bioink solution for core bio-printing. For 3D printing of the outer shell, EVs (3 × 10^11^, dissolved in PBS) were added to either DDW or AlgS (in DDW, 338 µg; 1:50,000 molar ratio) and incubated for 1.5 h at 37 °C, to allow equilibrium binding. Shell bioink solution was prepared by adding either EVs or AlgS-EVs complexes to the partially crosslinked alginate as described in 5.4.1. Shell bioink was printed at 16 °C, 0.3 bar and 18 mm/s printing velocity. After printing, the constructs were released from FRESH supporting bath by 1 h incubation at 37 °C, 5% CO_2_. The constructs were further crosslinked with 0.1 M CaCl_2_ (in DMEM without serum, 1% Pen-Strep) for 10 min, then washed once with release medium (DMEM without serum, 1% Pen-Strep).

For release studies, each patch was incubated in 1 mL release medium and incubated on a rotating incubator, at 37 °C, 5% CO_2_. The medium was collected and replaced every 2 days. Prior to EV measurement, each sample was centrifuged for 10 min at 6000 g, 4 °C. EV-containing supernatant was collected and the amount of EVs released into the media was determined by NTA.

### 5.9. Cell Metabolic Activity and DNA Content

Cell metabolic activity was evaluated by PrestoBlue reagent (Invitrogen). The 3D-bioprinted constructs were transferred into new wells and incubated with 600 µL of PrestoBlue reagent mixture (1:9 in DMEM culture medium supplemented with 20% FBS (*v*/*v*)) for 3 h at 37 °C. Aliquots (300 µL) were placed in a 96-well plate and the fluorescence was measured at excitation wavelength of 560 nm and emission wavelength of 590 nm using a plate reader (model ELX 808, BIO TEK Instruments, Winooski, VT). Samples of the reagent mixture was incubated under the same conditions without cells were used as a blank.DNA content was analyzed using the fluorescent dye bisbenzimidazole Hoechst 33258 (Sigma). the constructs were dissolved in 500 µL of sodium citrate (4% *w*/*v* in PBS) to free the cells. Cell suspensions were then centrifuged (6000 rpm, 10 min), and the cell pellet was suspended in 100 µL of lysis buffer (SDS 0.2%, *v*/*v*) in sodium citrate (0.015 M-saline, pH7) and incubated for 1 h at 37 °C. Then 100 µL of Hoechst 33258 assay solution (2 mg/mL) was added, followed by a 10-min incubation. Aliquots (180 µL) were placed in a 96-well plate, and the fluorescence was measured at an excitation wavelength of 485 nm and an emission wavelength of 530 nm using a plate reader. Samples of the reagent mixture incubated under the same conditions without cells were used as a blank. Specific metabolic activity was calculated as the cell metabolic intensity normalized to the DNA content quantified by Hoechst.

### 5.10. Immunostaining and Confocal Imaging

For immunofluorescence of 3D bio-printed structures, constructs were fixed in 4% (*v*/*v*) warm methanol free formaldehyde in DMEM buffer (1.8 mM CaCl_2_·H_2_O, 5.36 mM KCl, 0.81 mM MgSO_4_·7H_2_O, 0.1 M NaCl, 0.44 mM NaHCO_3_ and 0.9 Mm NaH_2_PO_4_) pH 7.4 for 20 min, washed 3 times in DMEM buffer, and permeabilized using Triton-X 100 (0.2% *v*/*v* in DMEM buffer) for 15 min at RT. The samples were washed 3 times, blocked for 1 h at RT in DMEM containing 1% BSA. The samples were incubated overnight with the primary antibodies, followed by 2-h incubation with secondary antibodies, with 3 × 10 min PBS washing between.

Alexa-Fluor 546-conjugated phalloidin (A22283, 1:1000, Life Technologies) was used for staining F-actin and NucBlue (Invitrogen) for nuclei detection. Imaging was performed with a Nikon C1si laser-scanning confocal microscope (LSCM). For reconstruction of 3D constructs, images were taken in 150 µm intervals between planes.

For quantification of cells 3D organization, cells perimeters were identified and the aspect ratio of each region of interest (ROI) was calculated based on F-actin staining, using imageJ thresholding and particle analysis of ROIs larger than 200 µm^2^.

### 5.11. Statistical Analysis

Statistical analysis was performed with GraphPad Prism version 8.43 for Windows (GraphPad Software, San Diego, CA, USA). All variables are expressed as mean ± SEM from at least 3 independent experiments, unless specified otherwise. Comparisons between two groups were performed by two-tailed Student *t*-test. For comparison between multiple groups, one-way analysis of variances (ANOVA) was performed with post hoc testing. For comparisons between two parameters, two-way ANOVA was performed with post hoc testing. Tukey’s correction was used to account multiple comparisons. Three-dimensional morphologies over time were compared using Kruskal–Wallis test with Dunn’s post hoc test. *p* < 0.05 was considered statistically significant unless noted otherwise.

## Figures and Tables

**Figure 1 gels-08-00769-f001:**
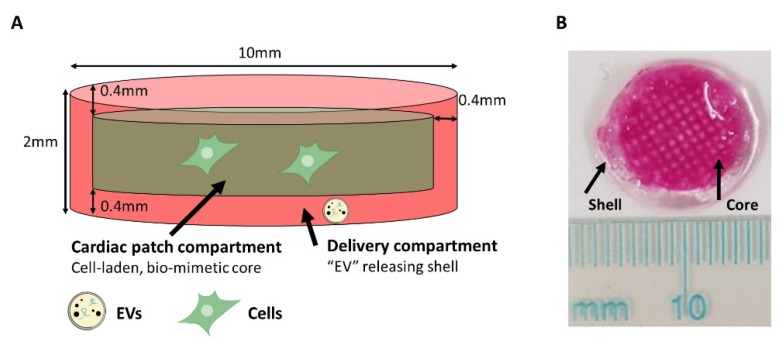
Cardiac patch design. (**A**) Schematic description of the designed cardiac patch. The inner core contains the cardiac cells; the outer shell contains EVs, affinity bound with AlgS, for EV delivery purposes. (**B**) Image of the EV-containing cardiac patches, fabricated by 3D printing. Polystyrene beads were used instead of cells for visualization purposes of the inner core.

**Figure 2 gels-08-00769-f002:**
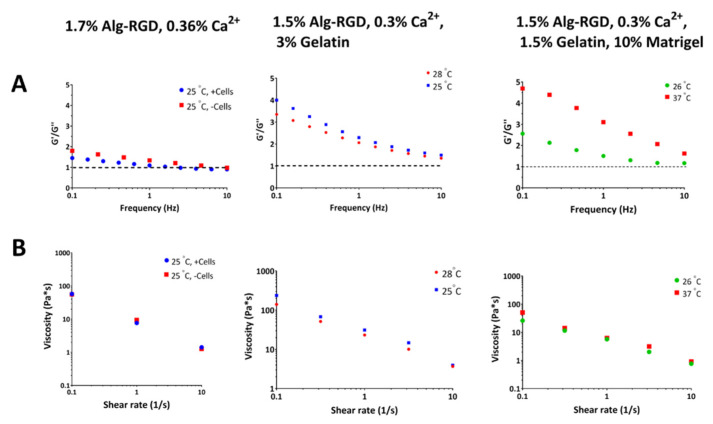
Viscoelastic responses of bioink solutions for cell-laden core 3D bio-printing. (**A**) Representative mechanical spectra of different core bioink solutions at different temperatures. The small-deformation oscillatory measurements are presented in terms of the storage modulus G′ (elastic response) and the loss modulus G″ (viscous response), as a function of angular frequency. G′ is used as the primary indicator of a gel-like (structured) system. In crosslinked unmodified alginate, at low angular frequencies, G′ values exceed G″ and with the increase in frequency, a cross-point appears. This is typical of an “entanglement network” type of chain interactions. (**B**) viscosity (bottom row) of different core bioink solutions at different temperatures.

**Figure 3 gels-08-00769-f003:**
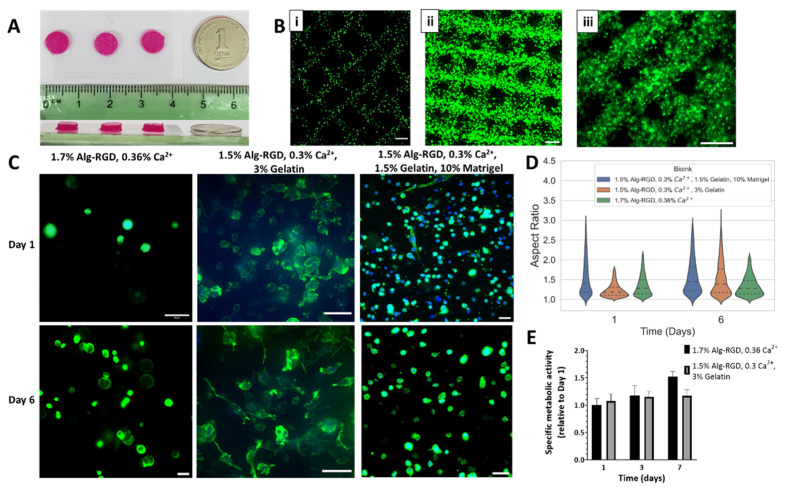
Three-dimensional bio-printing optimization of cardiac patch core. (**A**) Images of Alg-RGD cell-laden constructs, fabricated by 3D bio-printing. (**B**) Representative confocal images of 3D bio-printed strands using cell-laden bioink solutions, showing GFP-expressing human fibroblasts (HF-GFP) distribution (in green). (i) 1.7% Alg-RGD, 0.36% Ca^2+^; (ii) 1.5% Alg-RGD, 0.3% Ca^2+^, 3% gelatin; (iii) 1.5% Alg-RGD, 0.3% Ca^2+^, 1.5% gelatin, 10% Matrigel. Scale bar: 500 µm. (**C**) High magnification images of 3D bio-printed HF-GPF encapsulated in different bioink solutions, at day 1 (top) and 6 (bottom) post-printing. Nuclei: blue. F-actin: green. Scale bar: 50 µm. (**D**) Quantitative image analysis of cells’ aspect ratio distribution, printed within different bioink solution. Data represented in quartiles. (**E**) Specific metabolic activity of printed HF-GFP within 3D bio-printed constructs over time.

**Figure 4 gels-08-00769-f004:**
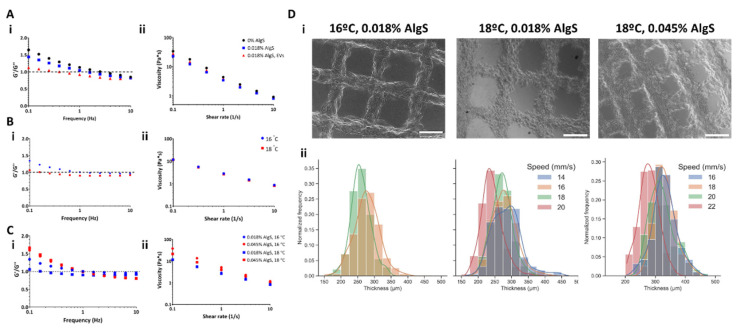
Three-dimensional bioprinting of alginate-based shell bioink solutions. (**A**,**B**) Mechanical spectra (i) and viscosity (ii) of 1.5% Alg, 0.36% Ca-gluconate (*w*/*v*, black) solutions without any additives, with 0.018% AlgS (*w*/*v*, blue) and 0.018% AlgS combined with 5 × 10^10^ EVs/mL (red). In 1.5% Alg, 0.36% Ca-gluconate (*w*/*v*) bioink solutions, cross-point appears at lower frequencies with addition of AlgS and EVs (red dashed lines). (**B**) Mechanical spectra (i) and viscosity (ii) of 1.5% Alg, 0.36% Ca-gluconate (*w*/*v*), 0.018% AlgS (*w*/*v*) solutions in 16 (blue) and 18 °C (red). (**C**) Mechanical spectra (i) and viscosity (ii) of 1.5% Alg, 0.36% Ca-gluconate (*w*/*v*), 0.018% (blue) and 0.045% (red) AlgS content (*w*/*v*). (**D**) Printing analysis of shell bioink solutions. (i) Images of 3D bio-printed grid patterns and (ii) strand thickness distribution plots, printed at different velocities, fabricated using 0.018% shell bioink printed at 16 (left) and 18 °C (middle), and 0.045% shell bioink printed at 18 °C (right). Scale bar: 500 µm.

**Figure 5 gels-08-00769-f005:**
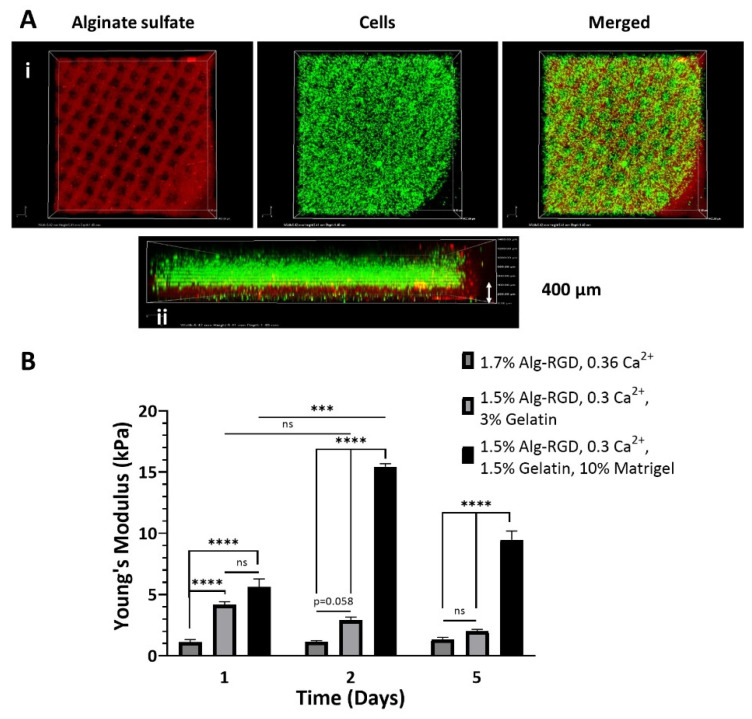
Three-dimensional bio-printing of cardiac patches. (**A**) Confocal images of XY (i) and reconstructed YZ (ii) planes of the two-compartment, 3D bio-printed cardiac patch. Red—Alginate sulfate; Green—fluorescently labeled cells. (**B**) Young’s modulus of 3D bio-printed cardiac patches over time, using different core bioink. Data are mean  ±  SEM, n = 3–4, *** *p* < 0.001, **** *p* < 0.0001, Tukey’s test, two-way ANOVA.

**Figure 6 gels-08-00769-f006:**
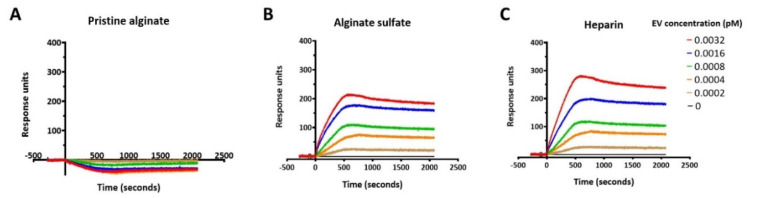
AlgS interacts with EVs through affinity binding. (**A**–**C**) Representative sensograms of different EVs concentrations with covalently bound pristine alginate (**A**), AlgS (**B**) and heparin (**C**) ligands.

**Figure 7 gels-08-00769-f007:**
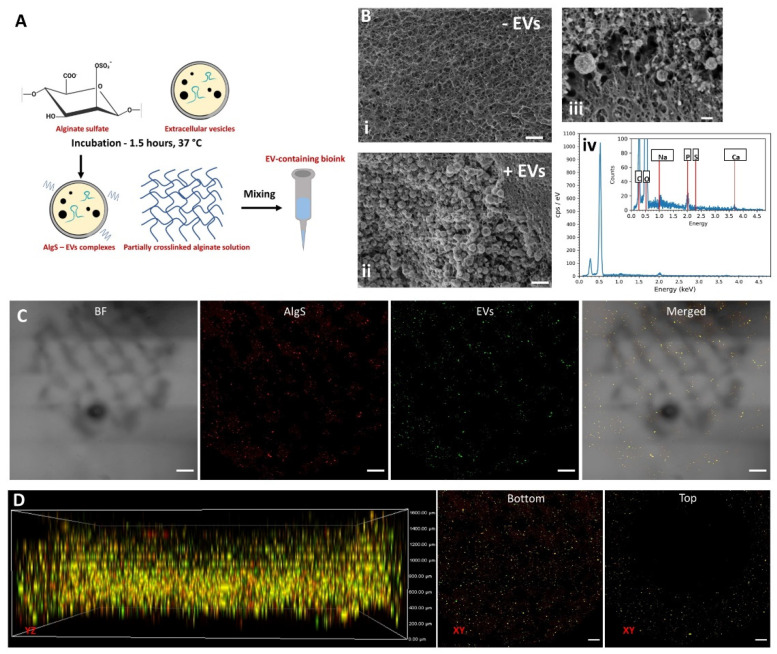
Three-dimensional bioprinting of AlgS-EV complexes. (**A**) A scheme describing the fabrication of alginate bioink with affinity bound EVs. Extracellular vesicles were reacted with alginate-sulfate and the bioconjugate solution was mixed with calcium cross-linked alginate solution. (**B**) Bioprinting of EVs within alginate bioink. Representative cryo-SEM images of alginate bioink constructs without EVs included (i) and with EV-AlgS complexes (ii). Scale bar: 200 nm. (iii) High magnification of alginate-containing 3D printed bioink. Scale bar: 100 nm. (iv) Representative EDX profile of EV-containing regions. (**C**,**D**) Confocal imaging of printed cardiac patches, demonstrating the locations of the different compartments. (**C**) Representative XY plane of EV-AlgS containing shell compartment. Scale bar: 500 µm. (**D**) 3D projections of printed constructs (left) and XY plane projections of bottom (middle) and top (right) layers of printed constructs. At the top layers, EV-AlgS complexes are located only at the outer shell, while at the lower, delivery compartment, EV-AlgS complexes are located within the entire printed area. CFSE-labeled EVs are in green, Cy5-labeled AlgS is in red. Scale bar: 500 µm.

**Figure 8 gels-08-00769-f008:**
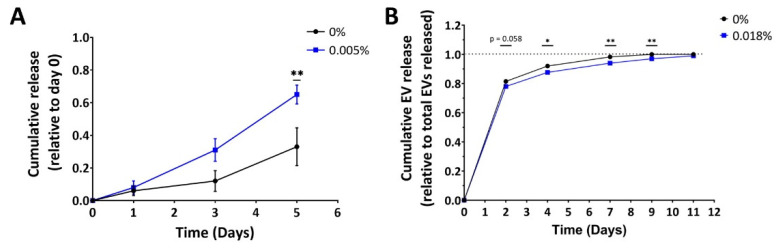
Alginate sulfate prolongs EV release from 3D-bioprinted alginate constructs. (**A**) Cumulative release from alginate beads (1.35% Alg) of EVs alone (blue) or incubated with alginate sulfate (0.005% *w*/*v*, 0.37% of total AlgS, purple), for the first 5 days. Each point represents the mean ± SEM cumulative release, relative to the total number of EVs on day 0. (**B**) Cumulative release from 3D printed alginate constructs (1.5% Alg, 0.36% Ca^2+^) of EVs alone (blue) or incubated with alginate sulfate (0.018% *w*/*v*, purple), for the first 11 days post-printing, relative to the total number of EVs released. Each point represents the mean ± SEM (n = 3–4) cumulative release, * *p* < 0.05, ** *p* < 0.01, compared by two-tailed Student *t*-test at each time point.

**Table 1 gels-08-00769-t001:** Final composition of cell-laden, core bioink solutions.

Bioink	Alg-RGD	Calcium Gluconate	Gelatin Type B	Matrigel™
1	1.7	0.36	0	0
2	1.5	0.3	3	0
3	1.5	0.3	1.5	10

All the components are detailed as weight % (*w*/*v*).

**Table 2 gels-08-00769-t002:** Printing parameters of cell-laden, core bioink solutions.

Bioink Solution #	Printing Parameters		
	Temperature (°C)	Pressure (bar)	Printing Velocity (mm/s)
1	20–22	0.6	20–22
2	28	0.5	12
3	26	0.5	18

## Data Availability

Not applicable.

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
