# Peer review of "Three-Dimensional Bio-Printed Cardiac Patch for Sustained Delivery of Extracellular Vesicles from the Interface"

_gels, 2022, doi:10.3390/gels8120769_

Round 1

Reviewer 1 Report

Cardiac tissue engineering has emerged as a promising strategy to treat infarcted cardiac tissues by replacing the injured region with ex vivo fabricated functional cardiac patch. Nevertheless, integration of the transplanted patch with host tissue is still a burden, limiting its clinical application. In this article,Assaf Bar et al. proposed a novel, 3D bio-printed cardiac patch design, composed of two compartments; an inner cell-laden compartment, enveloped by an outer compartment containing EVs. It is postulated that controlled, sustained delivery of EVs through release from alginate-AlgS matrix, would promote cardiac regeneration and improve integration if applied locally at the interface between the patch and surrounding tissue. This CP design comprises great potential at promoting more efficient patch assimilation with the host. I recommend this article for our journal. However, there are some problems for authors to improve their article before acceptance.

1. Part “delivery compartment "EV" releasing shell” in Figure 1A contains a white circle, is it related to the release process? How does this process work?

2. Figure 7A should better represent the process of forming bioink reactions, reaction mechanisms or results, rather than simply depicting the process of making them.

3. There is insufficient support for the practical application of live animals in the article.

4. The discussion section needs to be described more scientifically. It is suggested to add the strengths and limitations of the study and future directions.

5. Do they have the future plan to promoting more efficient patch assimilation with the host? Please give one or two examples.

6. It is suggested to add articles entitled “Ran Liu et al. Synthesis of nanomedicine hydrogel microcapsules by droplet microfluidic process and their pH and temperature dependent release” “Chitosan as Functional Biomaterial for Designing Delivery Systems in Cardiac Therapies” to the literature review.

7. Some mistakes in writing format need to be checked out carefully.

1) At line 130, “2.1.1 Cell-laden core bioink solutions” should be revised to “2.2.1 Cell-laden core bioink solutions”.

2) At line 197, “2.1.2 Core 3D bio-printing optimization” might be corrected to “2.2.2 Core 3D bio-printing optimization”.

3) Figures such as 3D are blurred and it is difficult to identify the information.

4) At line 709, “5.4” might be corrected to “5.5”.

5) At line 882, an extra space was typed.

Reviewer 2 Report

The manuscript by Bar et al reports the three-dimensional bio-printed cardiac patch for sustained delivery of extracellular vesicles from the interface.

1.     The manuscript needs moderate English spelling editing and grammar checks.

2.     Authors should review the text and correct any editing (ml) errors.

3.     The molecular weights of materials should be added to the materials section of the article.

4.     How many µm the scale bar lengths are is not clearly visible in Figure 3C. Figure 3 (B and C) should be added and rearranged.

5.     The result and discussion part can be strengthened with more references.

6.     The conclusion portion needs improvement.
